# Phagocyte Chemoattraction Is Induced through the Mcp-1–Ccr2 Axis during Efferocytosis

**DOI:** 10.3390/cells10113115

**Published:** 2021-11-10

**Authors:** Sang-Ah Lee, Deokhwan Kim, Chanhyuk Min, Byeongjin Moon, Juyeon Lee, Hyunji Moon, Susumin Yang, Chang Sup Lee, Gwangrog Lee, Daeho Park

**Affiliations:** 1School of Life Sciences, Gwangju Institute of Science and Technology, Gwangju 61005, Korea; sanga03@gist.ac.kr (S.-A.L.); po7322@gist.ac.kr (D.K.); alscksgur@gist.ac.kr (C.M.); byeongjinmoon@gist.ac.kr (B.M.); iris260@gist.ac.kr (J.L.); hjmoon311@gist.ac.kr (H.M.); susuminy@gist.ac.kr (S.Y.); gwangroglee@gist.ac.kr (G.L.); 2Laboratory of Cell Mechanobiology, Gwangju Institute of Science and Technology, Gwangju 61005, Korea; 3College of Pharmacy and Research Institute of Pharmaceutical Sciences, Gyeongsang National University, Jinju 52828, Korea; changsup@gnu.ac.kr

**Keywords:** efferocytosis, apoptotic cells, phagocytes, migration, chemoattraction, Mcp-1, Ccr2

## Abstract

Apoptotic cells generated during development and for tissue homeostasis are swiftly and continuously removed by phagocytes via a process called efferocytosis. Efficient efferocytosis can be achieved via transcriptional modulation in phagocytes that have engulfed apoptotic cells. However, such modulation and its effect on efferocytosis are not completely understood. Here, we report that phagocytes are recruited to apoptotic cells being cleared through the Mcp-1–Ccr2 axis, which facilitates clearance of apoptotic cells. We identified *Mcp-1* as a modulated transcript using a microarray and found that Mcp-1 secretion was augmented in phagocytes engulfing apoptotic cells. This augmented Mcp-1 secretion was impaired by blocking phagolysosomal degradation of apoptotic cells. Conditioned medium from wild type (WT) phagocytes promoted cell migration, but that from *Mcp-1^−/−^* phagocytes did not. In addition, blockade of Ccr2, the receptor for Mcp-1, abrogated cell migration to conditioned medium from phagocytes incubated with apoptotic cells. The intrinsic efferocytosis activity of *Mcp-1^−^*^/*−*^ and *Ccr2^−^*^/*−*^ phagocytes was unaltered, but clearance of apoptotic cells was less efficient in the peritoneum of *Mcp-1^−^*^/*−*^ and *Ccr2^−^*^/*−*^ mice than in that of *WT* mice because fewer Ccr2-positive phagocytes were recruited. Taken together, our findings demonstrate a mechanism by which not only apoptotic cells but also phagocytes induce chemoattraction to recruit phagocytes to sites where apoptotic cells are cleared for efficient efferocytosis.

## 1. Introduction

Apoptotic cells generated during development and for tissue homeostasis are swiftly and continuously removed through a process called efferocytosis [1,2]. Efficient efferocytosis is dependent on both phagocytes and apoptotic cells. Apoptotic cells do not passively wait to be cleared, but actively recruit phagocytes by secreting chemoattractants called ‘find-me’ signals such as nucleotides and modulate gene programs in the neighboring cells within a tissue by releasing metabolites as ‘goodbye’ signals [3,4,5]. In addition, apoptotic cells expose ‘eat me’ signals and simultaneously blunt ‘do-not-eat-me’ signals to be specifically engulfed [6,7]. The best-known eat-me signal is phosphatidylserine (PS), which is generally located in the inner leaflet of the plasma membrane in normal cells but translocates to the outer leaflet of the plasma membrane in apoptotic cells [8]. By contrast, CD47, a ‘do-not-eat-me’ signal, binds to SIRPα on phagocytes and disables phagocytosis [9,10,11]. However, CD47 clustering is disrupted in apoptotic cells and the interaction of CD47 with SIRPα is weakened [12]. Thus, these regulated eat-me and do-not-eat-me signals enable phagocytes to specifically engulf apoptotic cells. Phagocytes also express corresponding receptors for these signals and thus distinguish cells to be engulfed from those not to be engulfed [11]. For example, Tim-4 and Mertk, which are well-characterized engulfment receptors for apoptotic cells, directly recognize apoptotic cells by binding to PS on these cells and indirectly recognize apoptotic cells via bridging molecules such as Gas 6 [13,14,15,16]. Thus, phagocytes can sense cells to be cleared through interactions between PS and its receptors.

During efferocytosis, the transcriptional and translational programs of phagocytes are modulated for efficient clearance of apoptotic cells after they are engulfed and/or recognized. Transcription of genes involved in various processes is regulated in phagocytes, which increases the competence of phagocytes to recognize and engulf apoptotic cells promptly, and recruits neutrophils to promote digestion of ingested apoptotic cells [17,18,19,20]. Additionally, the levels of some proteins are translationally or post-translationally modulated in phagocytes for efficient efferocytosis after engulfment of apoptotic cells. For example, Ucp-2, Drp-1, and Orai1, which are involved in proton gradient dissipation in mitochondria and calcium flux, are upregulated in phagocytes engulfing apoptotic cells, causing phagocytes to engulf multiple apoptotic cells continuously [21,22,23].

The genes involved in efferocytosis and its signaling pathways have been identified over the past decades. Consequently, considerable progress has been made, especially with regard to molecules involved in recognition and ingestion of apoptotic cells. However, less is known about genes that are regulated in phagocytes during efferocytosis. Therefore, in this study, we performed a microarray to identify transcriptionally regulated genes and to evaluate their effects on efferocytosis. *Mcp-1* was transcriptionally modulated during efferocytosis. It was upregulated in phagocytes incubated with apoptotic cells and this was dependent on degradation of apoptotic cells in phagocytes. Upregulation of *Mcp-1* in phagocytes augmented secretion of Mcp-1, which promoted phagocyte chemoattraction. Depletion of Mcp-1 or Ccr2, the receptor for Mcp-1, in phagocytes did not alter their intrinsic efferocytosis activity; however, clearance of apoptotic cells was less efficient in *Mcp-1^−^*^/*−*^ and *Ccr2^−^*^/*−*^ mice due to a defect in phagocyte chemoattraction. Taken together, our findings imply that more phagocytes are recruited to apoptotic cells being engulfed through the Mcp-1-Ccr2 axis and that this facilitates clearance of apoptotic cells.

## 2. Materials and Methods

### 2.1. Reagents

The antibodies used in this study were anti-Ccr2 antibody ((NBP1-48338, Novus, St Charles, MO, USA), normal rabbit IgG (0111-01, SouthernBiotech, Birmingham, AL, USA), anti-CD16/32 antibody (#101302, BioLegend, San Diego, CA, USA), Goat anti-rabbit Alexa Fluor 488 (A11008, Invitrogen, Waltham, MA, USA), PE anti-mouse F4/80 (#123110, BioLegend, USA) and FITC anti-mouse F4/80 (#123108, BioLegend, USA).

Inhibitors and other reagents used in this study were Cytochalasin D (C8273-1MG, Sigma-Aldrich, Burlington, MA, USA), bafilomycin A1 (ab120497, Abcam, Waltham, MA, USA), apyrase (M0398S, NEB, Ipswich, MA, USA), PS beads (P-BOPS, Echelon Biosciences, Salt Lake City, UT, USA), TAMRA-SE (C1171, Life Technologies, Waltham, MA, USA), Dexamethasone (D1756-500MG, Sigma-Aldrich, USA), CellTracker^TM^ Green CMFDA dye (C7025, Thermo Fisher Waltham, MA, USA).

### 2.2. Mice

C57BL/6 were purchased from Taconic bioscience. *Tim-4^−^*^/*−*^ mice (RBRC04895) were obtained from Riken BioResource Center (Tsukuba, Japan). *Mertk^−^*^/*−*^ and *Ccr2^RFP^* mice were purchased from Jackson Laboratory, USA. *Mcp-1^−^*^/*−*^ mice were a generous gift from SungHoon Back at University of Ulsan. All mice were maintained and housed in equipped animal facility with temperature at 20–25 °C and humidity at 30–70%, under the same dark/light cycle (12:12). All experiments using mice were approved by the animal care and ethics committees of GIST in accordance with the national institutes of health guide for the care and use of laboratory animals.

### 2.3. Quantitative PCR

Total RNA from BMDMs incubated with or without apoptotic thymocytes was extracted using RNeasy Plus Mini Kit (74136, Qiagen, Dusseldorf, Germany). After that, cDNA was synthesized with SuperScript^®^ III First-Strand Synthesis System (18080-051, Invitrogen, Waltham, MA, USA). The levels of the indicated transcripts were measured using StepOnePlus real-time PCR system (Applied Biosystem, Waltham, MA, USA).

### 2.4. Preparation of Apoptotic Cells

Apoptotic Jurkat cells were prepared by irradiating Jurkat cells in DPBS with 100 mJ/cm^2^ ultraviolet-C (UVC). The equal volume of complete RPMI medium was immediately added and the cells were further incubated at 37 °C for 2 h. Apoptotic thymocytes were prepared as previously described. Briefly, thymocytes derived from 4- to 6-week-old C57BL/6 mice were stained with 50 μM TAMRA-SE or 0.5 μM CellTracker for 30 min and washed with RPMI medium containing 10% serum. Then, apoptosis in thymocytes was induced using 50 μM dexamethasone at 37 °C for 4 h in a 5% CO_2_ incubator.

### 2.5. ELISA

Peritoneal macrophages derived from the indicated mice or BMDMs were incubated with apoptotic Jurkat cells, PS beads, or PS liposomes in the absence or presence of Mfge8^D89E^ (10 μg/mL), cytochalasin D, or bafilomycin A1 for 8 h. Conditioned medium from the cells was collected, and Mcp-1 or Mcp-3 was measured using ELISA kit (900-K126 and 900-K123, PeproTech, Cranbury, NJ, USA) and ABTS ELISA buffer kit (900-K00, PeproTech, Cranbury, NJ, USA) according to the manufacturer’s protocol. Briefly, a 96-well plate was coated with an Mcp-1- or Mcp-3-capturing antibody for 12 h and blocked by blocking solution containing 1% BSA in PBS. One-hundred microliters of conditioned medium was added to the plate. After that, a detection antibody—avidin-HRP—and ABTS substrate were sequentially added and incubated for 2 h, 30 min, and 30 min, respectively. Fluorescence was measured using a microplate reader (VersaMax, Molecular Devices, San Jose, CA, USA).

### 2.6. Efferocytosis Assay

Efferocytosis assay was performed as previously described [24]. Basically, peritoneal macrophages derived from the indicated mice were incubated with TAMRA-labeled (C1171, Life Technologies, USA) apoptotic thymocytes for 15 min. A ratio of 1:10 (phagocytes to apoptotic cells) was used. To validate the effects of cytochalasin D, bafilomycin A1, or Mcp-1 on efferocytosis, the phagocytes were incubated with TAMRA-stained apoptotic thymocytes in the presence or absence of cytochalasin D for 8 h, bafilomycin A1 for 8 h, or Mcp-1 for 15 min. After that, the phagocytes were extensively washed with ice-cold PBS, trypsinized, and analyzed by flow cytometry (BD FACS Canto II, Franklin Lakes, NJ, USA). For in vivo efferocytosis assay, 1 × 10^7^ of TAMRA-labeled apoptotic thymocytes in 300 μL PBS were intraperitoneally injected into 8–10-week-old mice. Two hours after injection, peritoneal exudates were collected and stained with an FITC-conjugated anti-F4/80 antibody and analyzed by flow cytometry. TAMRA positive and F4/80 negative cells were considered as unengulfed apoptotic cells which were counted using Sphero AccuCount particles (ACBP-50-10, Spherotech, Lake Forest, IL, USA).

### 2.7. Migration Assay

Transwell migration assays were performed by applying 100 μL of THP-1 cells at 2 × 10^6^/mL to the upper chamber as the conditioned medium in the lower chamber (600 μL) of 8 μm pore size transwells (35224, SPL, Pocheon, Korea) at 37 °C for 6 h. The number of migrated THP-1 cells was manually determined using hemocytometer and the percentage of migrated cells was calculated as the percentage of input cells. To disrupt the gradient of Mcp-1, recombinant Mcp-1 (50 ng/mL) was added to the upper chamber with THP-1 cells. 0.025 units/mL apyrase (M0398S, NEB, Ipswich, MA, USA) was used for 5 min at room temperature to remove nucleotides in conditioned medium, and an anti-CCR2 antibody (5 μg) was pre-incubated with THP-1 cells for 30 min.

### 2.8. Ccr2 Positive Cell Recruitment

Unstained apoptotic cells (1 × 10^7^) in 300 μL PBS were intraperitoneally injected into 8–10-week-old *WT* and *Mcp-1^−^*^/*−*^ or *Ccr2^+^*^/*−*^ and *Ccr2^−^*^/*−*^ Mice. Two hours after injection, peritoneal exudates were collected, stained with anti-Ccr2 and PE- or FITC-conjugated anti-F4/80 antibodies, and analyzed by flow cytometry. The relative number of Ccr2 or RFP positive cells were counted using Sphero AccuCount particles (ACBP-50-10, Spherotech, Lake Forest, IL, USA).

### 2.9. Statistical Analysis

All data are shown as mean ± standard deviation. Each experiment was repeated at least three times independently. Data were analyzed using the GraphPad Prism 7 software (Prism 7, GraphPad Software, La Jolla, CA, USA). It was considered that differences were statistically significant when the *p*-values were less than 0.05.

## 3. Results

### 3.1. Phagocytes Release Mcp-1 during Engulfment of Apoptotic Cells

To identify genes that are modulated during efferocytosis, we performed a microarray in which RNA from bone marrow-derived macrophages (BMDMs) incubated with apoptotic thymocytes was compared with that from BMDMs not incubated with apoptotic thymocytes. In total, 283 genes were upregulated, and 80 genes were downregulated more than 1.5-fold in BMDMs incubated with apoptotic cells compared with control BMDMs (Figure 1a). Subsequent analysis revealed that expression of genes linked to cell differentiation, cell migration, cell proliferation, and immune responses was modulated in phagocytes incubated with apoptotic cells (Figure 1b). We decided to study the functional relevance of genes belonging to the chemokine family, which is related to cell migration, for efferocytosis because their function in this process is unknown. Three genes—*Mcp-1*, *Mcp-3*, and *Cxcl2*—were selected because they were upregulated to a relatively large extent in the microarray analysis and their transcripts were undetectable in thymocytes (Appendix A). Quantitative PCR analysis confirmed the transcriptional changes of *Mcp-1* and *Mcp-3*, but not of *Cxcl2*, detected in the microarray (Figure 1c). These two genes were also upregulated in peritoneal macrophages incubated with apoptotic cells. Moreover, they were upregulated more in peritoneal macrophages than in BMDMs (Figure 1d). Notably, expression of *Mcp-1* and *Mcp-3* was only marginally detected in apoptotic thymocytes, suggesting that these two genes are upregulated specifically in phagocytes. We next tested whether the increased levels of these two transcripts lead to increases of Mcp-1 and Mcp-3 at the protein level using an enzyme-linked immunosorbent assay (ELISA). Unexpectedly, only the level of Mcp-1, not of Mcp-3, was appreciably higher in BMDMs incubated with apoptotic cells than in control BMDMs (Figure 1e). Due to the distinctive upregulation of *Mcp-1* and *Mcp-3* in peritoneal macrophages incubated with apoptotic cells, we also investigated whether the levels of Mcp-1 and Mcp-3 are increased in peritoneal macrophages incubated with apoptotic cells. Mcp-1 was secreted by peritoneal macrophages in the basal state, whereas Mcp-3 was undetectable. When peritoneal macrophages were incubated with apoptotic cells, secretion of both Mcp-1 and Mcp-3 appreciably increased, and 10-fold more Mcp-1 than Mcp-3 was secreted (Figure 1f). These data imply that phagocytes release Mcp-1 and Mcp-3 during efferocytosis. Mcp-1 was significantly upregulated in both BMDMs and peritoneal macrophages at the transcript and protein levels, and phagocytes incubated with apoptotic cells produced much more Mcp-1 than Mcp-3; therefore, we focused mainly on Mcp-1 hereafter.

### 3.2. Phagolysosomal Acidification Is Necessary for Mcp-1 Secretion

Next, we investigated the mechanism by which secretion of Mcp-1 from phagocytes increases during efferocytosis. We first investigated whether a factor in the conditioned medium of apoptotic cells (apoptotic supernatants) stimulates secretion of Mcp-1. Mcp-1 secretion was not elevated by apoptotic supernatants but was robustly increased by apoptotic cells (Figure 2a and Appendix A), suggesting that apoptotic cells are crucial for release of Mcp-1 by phagocytes. Thus, we next investigated whether binding of apoptotic cells to phagocytes is important for Mcp-1 secretion. To this end, binding of apoptotic cells to phagocytes was blocked by Mfge8^D89E^, which binds to PS on apoptotic cells but not to integrins on phagocytes [25]. Treatment of apoptotic cells with Mfge8^D89E^ abolished not only efferocytosis, but also the elevation of Mcp-1 secretion by peritoneal macrophages (Figure 2b and Appendix A). In addition, peritoneal macrophages derived from *Tim-4^−^*^/*−*^ and *Mertk^−^*^/*−*^ mice secreted substantially less Mcp-1 than wild type (WT) controls when they were incubated with apoptotic cells (Figure 2c). These data imply that PS recognition is necessary for Mcp-1 secretion during efferocytosis. We next investigated whether PS recognition is sufficient for Mcp-1 secretion. To address this, we allowed phagocytes to bind to apoptotic cells, but not to internalize them, using cytochalasin D, an inhibitor of actin polymerization. Cytochalasin D reduced Mcp-1 secretion by peritoneal macrophages incubated with apoptotic cells in a dose-dependent manner, which was paralleled by a similar decrease in the percentage of phagocytes engulfing apoptotic cells (Figure 2d,e). This suggests that binding of apoptotic cells to phagocytes is insufficient to induce Mcp-1 secretion and that internalization of apoptotic cells or a subsequent step(s) is necessary for elevation of Mcp-1 secretion. The level of Mcp-1 secretion was commensurate with the percentage of phagocytes engulfing apoptotic cells, and PS recognition was necessary but not sufficient for elevation of Mcp-1 secretion. Therefore, degradation of apoptotic cells in phagocytes is likely crucial for elevation of Mcp-1 secretion. To investigate this, we blocked phagolysosomal degradation of apoptotic cells using bafilomycin A1, which inhibits acidification of phagolysosomes and thus degradation of phagolysosomal cargos. Bafilomycin A1 diminished Mcp-1 secretion by peritoneal macrophages incubated with apoptotic cells without affecting the efficiency of efferocytosis (Figure 2f,g), implying that degradation of apoptotic cells in phagocytes is required for elevation of Mcp-1 secretion during efferocytosis. This notion was supported by experiments using indigestible or simplified surrogates mimicking apoptotic cells, namely, PS beads and PS liposomes, respectively. Incubation with neither PS beads nor PS liposomes elevated Mcp-1 secretion by peritoneal macrophages (Figure 2h,i). In summary, these data suggest that phagolysosomal degradation of apoptotic cells is necessary for elevation of Mcp-1 secretion by phagocytes. In addition, they indicate that engulfment receptors for apoptotic cells are not directly linked to elevation of Mcp-1 secretion during efferocytosis and that *Tim-4^−^*^/*−*^ and *Mertk^−^*^/*−*^ peritoneal macrophages secrete less Mcp-1 because they contain fewer engulfed apoptotic cells than control phagocytes.

### 3.3. Mcp-1 Promotes Phagocyte Chemoattraction

Mcp-1 is a member of the CC chemokine family that mediates chemotaxis and induces monocyte chemoattraction [26]. Thus, we hypothesized that, in addition to find-me signals, Mcp-1 secreted by phagocytes engulfing apoptotic cells also functions as a chemoattractant to recruit more phagocytes to sites where apoptotic cells are being cleared. To test this, we first investigated whether Mcp-1 secreted by engulfing phagocytes induces phagocyte migration using a transwell migration assay. Conditioned medium from phagocytes incubated with apoptotic cells enhanced migration of THP-1 monocytes, whereas conditioned medium from *Mcp-1^−^*^/*−*^ peritoneal macrophages incubated with apoptotic cells did not (Figure 3a). Addition of purified Mcp-1 to the upper chamber of the transwell abrogated migration of THP-1 cells to the lower chamber (Figure 3b), suggesting that Mcp-1 in conditioned medium induces phagocyte migration. Noticeably, the enhanced migration of phagocytes was not caused by nucleotides such as ATP released from apoptotic cells that function as chemoattractants because apyrase, a ATP-diphosphatase, had no effect on cell migration induced by conditioned medium, and the level of ATP was much lower in conditioned medium than in apoptotic supernatants, which is likely due to degradation of ATP by CD39, an ectonucleotidase that catalyzes hydrolysis of triphosphonucleosides to the monophosphonucleoside derivative, expressed in peritoneal macrophages (Figure 3c–e) [27,28]. Ccr2 is the receptor for Mcp-1 and is highly expressed in monocytes [29]. Thus, we next investigated whether the enhanced migration of THP-1 cells to conditioned medium is dependent on Ccr2. To this end, we disabled Ccr2 using an anti-Ccr2 antibody and performed a transwell migration assay. Blockade of Ccr2 with an anti-Ccr2 antibody appreciably diminished migration of THP-1 cells to conditioned medium (Figure 3f). These data suggest that phagocyte migration is induced through the Mcp-1-Ccr2 axis during clearance of apoptotic cells.

### 3.4. Clearance of Apoptotic Cells Is Impaired in Mcp-1^−/−^ and Ccr2^−/−^ Mice

Next, to investigate the relevance of Mcp-1 to efferocytosis, we first evaluated its effects on efferocytosis. Efferocytosis by *Mcp-1**^−^*^/*−*^ peritoneal macrophages was similar to that by *WT* peritoneal macrophages, as indicated by the percentage and MFI (mean fluorescence intensity, representing the relative number of apoptotic cells per phagocyte) of phagocytes that engulfed apoptotic cells (Figure 4a). Moreover, addition of purified Mcp-1 did not affect the efficiency of efferocytosis by peritoneal macrophages (Figure 4b). These data imply that Mcp-1 is unrelated to the intrinsic efferocytosis capability.

A crucial feature of efferocytosis is recruitment of phagocytes to sites where apoptotic cells are generated to facilitate their clearance, which is achieved by find-me signals released from apoptotic cells. Mcp-1 was released from phagocytes during efferocytosis and enhanced phagocyte chemoattraction, and therefore may play a similar role as find-me signals to facilitate recruitment of phagocytes and thus clearance of apoptotic cells. Thus, we next investigated whether the inability of *Mcp-1**^−^*^/*−*^ phagocytes to induce cell migration is linked to a defect of apoptotic cell clearance in vivo, although the intrinsic efferocytosis ability of *Mcp-1**^−^*^/*−*^ peritoneal macrophages remains unchanged. To this end, apoptotic cells were intraperitoneally injected and unengulfed apoptotic cells were measured. There were more unengulfed apoptotic cells in the *Mcp-1**^−^*^/*−*^ peritoneum than in the *WT* peritoneum (Figure 4c and Appendix A). Interestingly, fewer Ccr2-positive cells were recruited to the *Mcp-1**^−^*^/*−*^ peritoneum than to the WT peritoneum (Figure 4d), which correlated with the number of unengulfed apoptotic cells, implying that recruitment of fewer phagocytes to the *Mcp-1**^−^*^/*−*^ peritoneum leads to an increase in unengulfed apoptotic cells. Furthermore, we validated the defect of apoptotic cell clearance in *Mcp-1**^−^*^/*−*^ mice using *Ccr2**^−^*^/*−*^ mice. Ccr2 deletion is indicated by RFP expression in cells expressing Ccr2. Using this feature, we evaluated recruitment of Ccr2-expressing cells to the peritoneum after intraperitoneal injection of apoptotic cells. Efferocytosis by *Ccr2**^−^*^/*−*^ peritoneal macrophages was comparable with that by *Ccr2^+^*^/*−*^ peritoneal macrophages (Figure 4e). However, there were more unengulfed apoptotic cells in the *Ccr2**^−^*^/*−*^ peritoneum than in the *Ccr2^+^*^/*−*^ peritoneum (Figure 4f and Appendix A). In addition, similar to *Mcp-1**^−^*^/*−*^ mice, fewer RFP-positive cells were recruited to the *Ccr2**^−^*^/*−*^ peritoneum than to the *Ccr2^+^*^/*−*^ peritoneum (Figure 4g), suggesting that phagocytes are recruited through the Mcp-1-Ccr2 axis during efferocytosis.

Collectively, transcriptional programs are modulated in phagocytes engulfing apoptotic cells, and this facilitates prompt and continuous clearance of apoptotic cells. Our observations demonstrate a mechanism by which this is achieved. Specifically, phagocyte chemoattraction is induced through the Mcp-1-Ccr2 axis during efferocytosis, resulting in recruitment of more phagocytes to apoptotic cells being phagocytosed and efficient clearance of these cells.

## 4. Discussion

Apoptotic cells are not passive cargos engulfed by phagocytes, but actively play roles for efficient efferocytosis. They secrete various molecules, including nucleotides, S1P, and lactoferrin, which induce chemotaxis [3,30,31,32]. These molecules recruit specific professional phagocytes such as monocytes and macrophages but repel undesired phagocytes such as neutrophils and help phagocytes find apoptotic cells promptly, which contributes to efficient clearance of apoptotic cells. In this study, we showed that secretion of Mcp-1 by phagocytes is induced during efferocytosis, and that this leads to recruitment of more phagocytes to sites where apoptotic cells are engulfed and thus enhances clearance of these cells. Thus, Mcp-1 functions similarly to find-me signals. Mcp-1 is released by phagocytes instead of apoptotic cells and can therefore be called a ‘help-me’ signal. Clearance of apoptotic cells was less efficient in *Ccr2^−/−^* mice than in *Mcp-1^−/−^* mice. This phenomenon may be due to the redundancy of chemokines released by phagocytes engulfing apoptotic cells. Secretion of Mcp-3, as well as Mcp-1, was elevated in phagocytes engulfing apoptotic cells (Figure 1f). In addition, phagocytes engulfing apoptotic cells may release other chemokines besides Mcp-1 and Mcp-3. Indeed, Ccr2 is a promiscuous receptor for a variety of chemokines including CC and CXC chemokines [33]. Thus, Ccr2 may affect phagocytic recruitment much more than a chemokine that binds to it, resulting in the more severe defect of apoptotic cell clearance in *Ccr2^−/−^* mice. It remains to be elucidated whether phagocytes engulfing apoptotic cells secrete other chemokines that bind to Ccr2 as a receptor.

A recent study showed that apoptotic cell-derived metabolites metabolized in phagocytes upregulate Dbl by stabilizing *Dbl* mRNA, leading to engulfment of multiple apoptotic cells by phagocytes [34]. In this study, we showed that indigestible or simplified surrogates of apoptotic cells, namely, PS beads and PS liposomes, respectively, failed to elevate Mcp-1 secretion. In addition, phagolysosomal degradation of apoptotic cells was necessary for elevation of Mcp-1 secretion by phagocytes. These findings indicate that metabolites derived from apoptotic cells likely regulate the level of *Mcp-1* and that upregulation of *Mcp-1* during efferocytosis can be due not only to transcriptional regulation but also alteration of *Mcp-1* stability.

In summary, our observations suggest that during efferocytosis, phagocytes release Mcp-1 to recruit more phagocytes to sites where apoptotic cells are being engulfed and thus facilitate clearance of these cells. Due to the correlation between defects in efferocytosis and autoimmune-related diseases, phagocyte chemoattraction through the Mcp-1-Ccr2 axis may have implications to develop therapeutics for these diseases.

## Figures and Tables

**Figure 1 cells-10-03115-f001:**
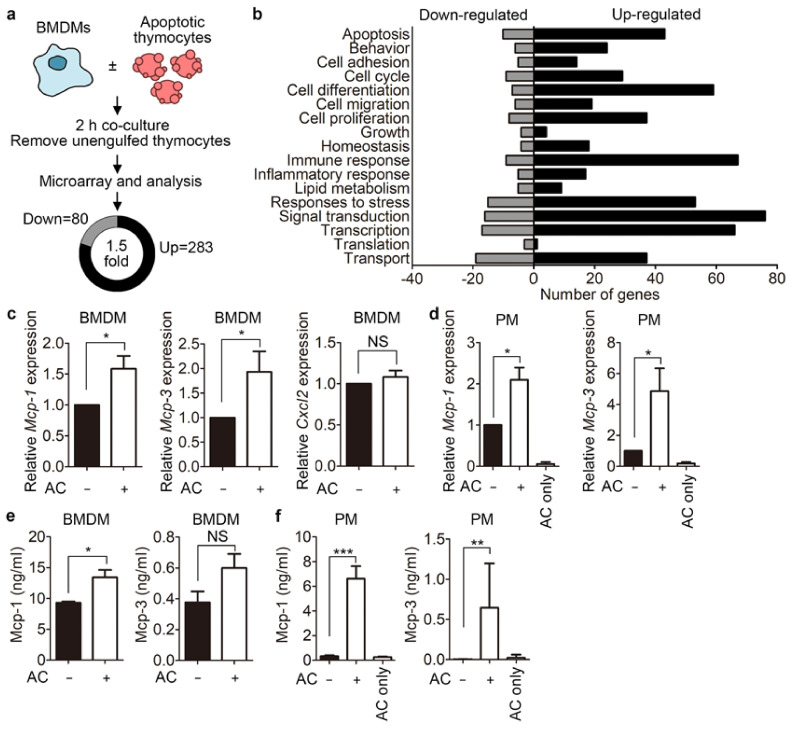
Mcp-1 secretion by phagocytes is augmented during efferocytosis. (**a**) Schematic diagram showing how genes regulated during efferocytosis were identified. BMDMs were incubated with or without apoptotic thymocytes for 2 h and then transcriptional changes were compared between these two samples. The numbers of up- and downregulated genes in phagocytes incubated with apoptotic cells compared with control phagocytes are shown. (**b**) Gene ontology analysis. Genes up- or downregulated more than 1.5-fold in phagocytes incubated with apoptotic cells compared with control phagocytes were categorized according to their function. BMDMs (**c**) or peritoneal macrophages (**d**) were incubated with or without apoptotic thymocytes for 2 h, and the transcript levels of *Mcp-1*, *Mcp-3*, and *Cxcl2* (c) or *Mcp-1* and *Mcp-3* (d) were measured using quantitative RT-PCR. BMDMs (**e**) or peritoneal macrophages (**f**) were incubated with or without apoptotic Jurkat for 8 h, and then conditioned medium from phagocytes was collected. The protein levels of Mcp-1 and Mcp-3 were measured using an ELISA. All data are shown as the mean ± SEM. * *p* < 0.05, ** *p* < 0.01, *** *p* < 0.001. NS, not significant; PM, peritoneal macrophages; AC, apoptotic cells.

**Figure 2 cells-10-03115-f002:**
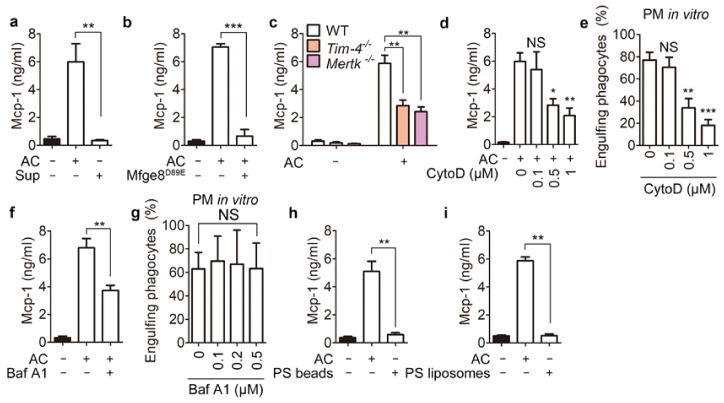
Degradation of apoptotic cells in phagocytes is necessary for Mcp-1 secretion. (**a**) Peritoneal macrophages were incubated with apoptotic cells or apoptotic supernatants for 8 h. The levels of Mcp-1 in conditioned medium from the cells were measured using an ELISA. (**b**) Peritoneal macrophages were incubated with apoptotic cells treated with or without Mfge8^D89E^ (10 μg/mL) for 8 h. The levels of Mcp-1 were measured as in panel (**a**). (**c**) Peritoneal macrophages derived from *WT*, *Tim-4^−^*^/*−*^, or *Mertk^−^*^/*−*^ mice were incubated with apoptotic cells for 8 h and then the levels of Mcp-1 in conditioned medium from the cells were measured using an ELISA. (**d**,**e**) Peritoneal macrophages were incubated with apoptotic cells in the presence of the indicated concentrations of cytochalasin D for 8 h. The levels of Mcp-1 in conditioned medium (**d**) and efferocytosis by the cells (**e**) were measured. Peritoneal macrophages were incubated with apoptotic cells in the presence of 0.5 μM (**f**) or the indicated concentrations (**g**) of bafilomycin A1. The levels of Mcp-1 in conditioned medium (**f**) and efferocytosis by the phagocytes (**g**) were evaluated. (**h**,**i**) Peritoneal macrophages were incubated with apoptotic cells, PS beads, or PS liposomes for 8 h. The levels of Mcp-1 in conditioned medium were measured as in a. All data are shown as the mean ± SEM. * *p* < 0.05, ** *p* < 0.01, *** *p* < 0.001. NS, not significant; PM, peritoneal macrophages; AC, apoptotic cells; Sup, supernatants; CytoD, cytochalasin D; Baf A1, bafilomycin A1; PS, phosphatidylserine.

**Figure 3 cells-10-03115-f003:**
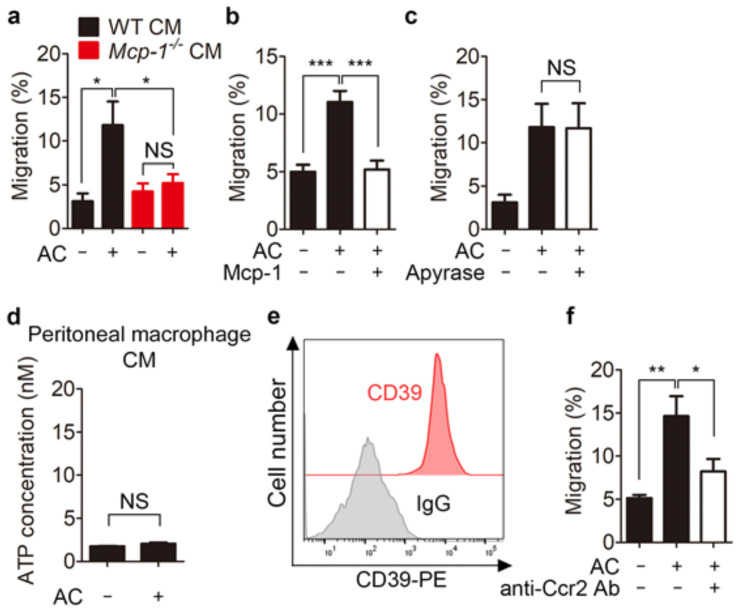
Conditioned medium from phagocytes during efferocytosis attracts monocytes in a Mcp-1-dependent manner. (**a**) Migration of THP-1 monocytes through a transwell to conditioned medium from *WT* or *Mcp-1^−^*^/*−*^ peritoneal macrophages incubated with or without apoptotic cells was measured. The percentages of input monocytes that migrated to the lower chamber are shown. (**b**) Pure Mcp-1 (50 ng/mL) was added to the upper well along with THP-1 cells, and migration was assessed toward conditioned medium from phagocytes incubated with apoptotic cells placed in the lower chamber. (**c**) Conditioned medium derived from phagocytes incubated with apoptotic cells was treated with or without apyrase, and then migration of THP-1 cells toward the conditioned medium was assessed. (**d**) ATP concentration in conditioned medium derived from phagocytes incubated with or without apoptotic cells were measured using a colorimetric method. (**e**) Peritoneal macrophages were stained with a PE-conjugated anti-CD39 antibody or isotype control antibody, and analyzed by flow cytometry. (**f**) THP-1 cells were pretreated with an anti-Ccr2 antibody. Migration of THP-1 cells toward conditioned medium derived from phagocytes incubated with apoptotic cells was evaluated as in a. All data are shown as the mean ± SEM. * *p* < 0.05, ** *p* < 0.01, *** *p* < 0.001. NS, not significant; CM, conditioned medium; AC, apoptotic cells; Ab, antibody.

**Figure 4 cells-10-03115-f004:**
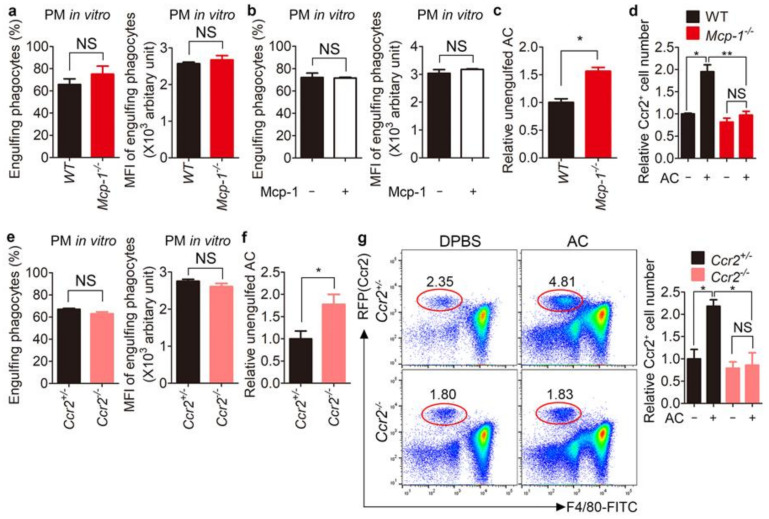
Clearance of apoptotic cells is impaired in *Mcp-1^−^*^/*−*^ and *Ccr2^−^*^/*−*^ mice. (**a**) Peritoneal macrophages derived from *WT* or *Mcp-1^−^*^/*−*^ mice were incubated with TAMRA-labeled apoptotic thymocytes for 15 min. The percentages and MFIs of phagocytes engulfing apoptotic cells were measured by flow cytometry. (**b**) Peritoneal macrophages were incubated with TAMRA-labeled apoptotic thymocytes in the absence or presence of pure Mcp-1 for 15 min. Phagocytes engulfing apoptotic cells were analyzed by flow cytometry. (**c**) TAMRA-labeled apoptotic thymocytes were intraperitoneally injected into *WT* or *Mcp-1^−^*^/*−*^ mice. Two hours after injection, peritoneal exudates were collected, stained with an anti-F4/80 antibody, and analyzed by flow cytometry. The relative number of TAMRA-positive and F4/80-negative cells per counting bead was determined. (**d**) Apoptotic thymocytes were intraperitoneally injected into WT or *Mcp-1^−^*^/*−*^ mice. Two hours after injection, peritoneal exudates were stained with an anti-Ccr2 antibody and analyzed by flow cytometry. The relative number of Ccr2-positive cells per counting bead was determined. (**e**) Peritoneal macrophages derived from *Ccr2^+^*^/*−*^ or *Ccr2^−^*^/*−*^ mice were incubated with CellTracker-labeled apoptotic thymocytes and analyzed by flow cytometry. (**f**) CellTracker-labeled apoptotic thymocytes were intraperitoneally injected into *Ccr2^+^*^/*−*^ or *Ccr2^−^*^/*−*^ mice. Two hours after injection, peritoneal exudates were stained with an anti-F4/80 antibody and analyzed by flow cytometry. The relative number of CellTracker-positive and F4/80-negative cells per counting bead was determined. (**g**) Apoptotic thymocytes were intraperitoneally injected into *Ccr2^+^*^/*−*^ or *Ccr2^−^*^/*−*^ mice. Two hours after injection, peritoneal exudates were collected and stained with an anti-F4/80 antibody. The relative number of RFP-positive cells per counting bead was determined. Red circles indicate RFP (Ccr2)-positive cells. All data are shown as the mean ± SEM. * *p* < 0.05, ** *p* < 0.01. NS, not significant; PM, peritoneal macrophages; AC, apoptotic cells.

## Data Availability

Data supporting the findings of the study are available within the article or from the corresponding author upon reasonable request.

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
