# Peer review of "Phagocyte Chemoattraction Is Induced through the Mcp-1–Ccr2 Axis during Efferocytosis"

_cells, 2021, doi:10.3390/cells10113115_

Round 1

Reviewer 1 Report

Overall, Lee et al aimed to investigate the functional impact of engulfment on the phagocytosing cell. They identified that upon engulfment, phagocytes can upregulate and secrete MCP-1 to further recruit phagocytes as a so-called ‘help-me’ signal. This is a great study that is well written, easy to read and relatively comprehensive. The combination of in vitro and in vivo assays, as well as genetically KO models complement one another well to support the authors conclusions.

Major comments:

The only major comment is to include more supportive data for Figure 4c and f/g, as these are the fundamental experiments supporting the entire conclusions of the paper. For example, authors should show the percentage of engulfing phagocytes in the peritoneum. Similarly, in this experiment, can authors show that the engulfment by local phagocytes is not impaired by Mcp-1-/- in vivo, but rather is it specifically the engulfment by recruited phagocytes that leads to an accumulation of apoptotic cells? (i.e. F4/80+ (local) or Ccr2+ (infiltrating) and TAMRA+ double positive cells). Moreover, figure 4c (and also f) should show absolute numbers of remaining apoptotic cells (rather than relative) so readers can gain a clearer understanding of the efficiency of this assay. Finally, it is surprising that the lack of recruited CCr2+ cells (Fig 4g) results in nearly double the amount of uncleared apoptotic cells (Fig 4f), when they only represent ~4.8% of the entire peritoneum, and F4/80+ macs are likely to have a predominate role in engulfment. Thus, more data to support these conclusions would be appreciated.

Minor comments

  • In the introduction, it would be worthwhile including the recent discovery of ‘goodbye signals’ where phagocytes can release a series of metabolites to modulate the surrounding environment. https://pubmed.ncbi.nlm.nih.gov/32238926/
  • Figure 1: This study is built on the initial micro RNA data which is presented and validated in Fig 1. However, for this experiment authors looked at total BMdMs that were incubated with apoptotic cells, rather that exclusively isolating the engulfing BMdMs. Engulfing phagocytes (and the presence of apoptotic cells alone) can modulate the surrounding environment. Thus, is it preferential to specifically isolate engulfing cells when performing such assays. At a minimum, authors should not denote these phagocytes as ‘engulfing’ (such as on line 164, and conclusions on line 187) but rather state that phagocytes are incubated with apoptotic cells.
  • In the supp data, authors need to show the validation of apoptosis induction in the Jurkats post UV (How much apoptosis and PS exposure actually is present by only 2 hs?).
  • It would be interesting to look at phagocytes that have engulfed multiple targets. Does a phagocyte that engulfed 3 apoptotic cells secrete more Mcl-1 than one who only engulfed 1? Or, is there no change?
  • Line 169-171: Authors should show the initial microRNA data for MCP-1, -3 and Cxcl2.
  • Figure 2b: authors should show/validate the reduction of engulfment caused by Mfge8 mutation, before showing the reduction in secretion. I.e, it would be of interest to know whether the completion inhibition of Mcp-1 secretion correlates with a total abolishment of engulfment (especially considering there are many receptors/molecules that contribute to engulfment). This is the same for Figure 2c.
  • Figure 3d: Is this the supernatant from the entire culture (phagocytes only, and phagocytes + apoptotic cells)? If so this data is concerning as there should significant levels of ATP release by the apoptotic cells? (especially given that ATP release is so well characterised in apoptotic Jurkats…). If no, this annotation/explanation needs to be clearer.
  • Figure 4a-b: Why were engulfment assays performed only for 15 min, when above and in figure 4f, they were performed for >2 hours? It is surprising to see such high levels of engulfment after only 15 min.. Also, why do these assays use TAMRA rather that a cell tracking dye like below? Please note that a better and more accurate system to use is staining target cells with CypHer-5E or pHrodo, which can specifically detect engulfment. It would be good to repeat some assays (such as Fig 4c) with this system.
  • Overall, addition annotations on the figures would be appreciated, indicating details such as the type of phagocyte, in vitro or in vivo engulfment etc. In all engulfment assays or at least in the methods, authors should clearly indicate the ratio of target cell: phagocyte used.

Reviewer 2 Report

In this manuscript, the authors described the ability of macrophages to release the chemokine Mcp-1 following engulfment and digestion of apoptotic cells. The release of Mcp-1 can subsequently recruit Ccr2+ phagocytes to aid further clearance of apoptotic cells. The proposed model is quite interesting, whereby macrophages can release the so called ‘help-me’ signal to aid prompt removal of apoptotic cells. Overall, the manuscript is very interesting and relevant to the field, providing ample of data to support their conclusions, including mechanistic insights. The authors should also consider the following minor points:

  1. It is premature to conclude “metabolites derived from apoptotic cells in phagocytes increase Mcp-1 secretion”? Please reword accordingly.
  2. In relation to data presented in Figure 3C, maybe it is better to conclude that apyrase had no effect rather than “only marginally inhibited cell migration”. Please reword accordingly.
  3. Regarding data presented in Figure 4C, out of interest, in the same experiment, what was the level of TAMRA-positive F4/80-positive cells when comparing WT and Mcp-1-/-?
  4. The concept of macrophage releasing chemokine following apoptotic cell uptake has been examined previously (e.g. Iyoda et al. Neutrophils accelerate macrophage-mediated digestion of apoptotic cells in vivo as well as in vitro. J Immuno 2005). Some references and further discussion on these previous studies are warranted.
